# Addressing the need for interactive, efficient, and reproducible data processing in ecology with the datacleanr R package

Alexander G. Hurley[1]*, Richard L. Peters[2,3], Christoforos Pappas[4,5,6], David N. Steger[1,7,8], Ingo Heinrich[1,7,8]

1 Climate Dynamics and Landscape Evolution, GFZ German Research Centre for Geosciences, Potsdam, Germany, 2 Laboratory of Plant Ecology, Department of Plants and Crops, Faculty of Bioscience Engineering, Ghent University, Ghent, Belgium, 3 Department of Environmental Sciences, University of Basel, Basel, Switzerland, 4 Centre d'étude de la forêt, Université du Québec à Montréal, Montréal, Canada, 5 Département Science et Technologie, Téluq, Université du Québec, Montréal, Canada, 6 Department of Civil Engineering, University of Patras, Rio Patras, Greece, 7 Humboldt-Universität zu Berlin, Berlin, Germany, 8 Natural Sciences Unit, German Archaeological Institute DAI, Berlin, Germany

* hurley@gfz-potsdam.de

**Data Availability Statement:** All software and data necessary for reproducibility are publicly available.

## Abstract

Ecological research, just as all Earth System Sciences, is becoming increasingly data-rich. Tools for processing of "big data" are continuously developed to meet corresponding technical and logistical challenges. However, even at smaller scales, data sets may be challenging when best practices in data exploration, quality control and reproducibility are to be met. This can occur when conventional methods, such as generating and assessing diagnostic visualizations or tables, become unfeasible due to time and practicality constraints. Interactive processing can alleviate this issue, and is increasingly utilized to ensure that large data sets are diligently handled. However, recent interactive tools rarely enable data manipulation, may not generate reproducible outputs, or are typically data/domain-specific. We developed datacleanr, an interactive tool that facilitates best practices in data exploration, quality control (e.g., outlier assessment) and flexible processing for multiple tabular data types, including time series and georeferenced data. The package is open-source, and based on the R programming language. A key functionality of datacleanr is the "reproducible recipe"—a translation of all interactive actions into R code, which can be integrated into existing analyses pipelines. This enables researchers experienced with script-based workflows to utilize the strengths of interactive processing without sacrificing their usual work style or functionalities from other (R) packages. We demonstrate the package's utility by addressing two common issues during data analyses, namely 1) identifying problematic structures and artefacts in hierarchically nested data, and 2) preventing excessive loss of data from 'coarse,' code-based filtering of time series. Ultimately, with datacleanr we aim to improve researchers' workflows and increase confidence in and reproducibility of their results.

The software is available from an online repository at https://github.com/the-Hull/datacleanr and via CRAN (https://cran.r-project.org/package= datacleanr); the software version used for this publication is archived at https://doi.org/10.5281/ zenodo.6337609. Eddy covariance data are available online from the FLUXNET2015 webpage (http://fluxnet.fluxdata.org/data/fluxnet2015-dataset/). The allometry and trait data is available at https://github.com/dfalster/baad. The Berlin street and park tree data is available at https://daten. berlin.de/. The data to reproduce the profiling and time series cleaning example are archived at https://doi.org/10.5281/zenodo.4550726.

**Funding:** AGH and IH were supported through the Helmholtz-Climate-Initiative (HI-CAM), funded by the Helmholtz Initiative and Networking Fund (https://www.helmholtz.de/en/about-us/the-association/initiating-and-networking/); the authors are responsible for the content of this publication. RLP acknowledges support of the Swiss National Science Foundation (http://www.snf.ch/), Grant P2BSP3_184475. The funders had no role in study design, data collection and analysis, decision to publish, or preparation of the manuscript.

**Competing interests:** The authors have declared that no competing interests exist.

## Introduction

Ecology, just as all Earth system sciences, is increasingly data-rich [e.g., 1]. These data are a boon for novel inferences, and increasingly inform decision making [2, 3], for example, through databases from coordinated efforts that facilitate synoptic studies of carbon fluxes [e.g., FLUXNET, 4] and stocks [5] or ecosystem functioning [e.g., via trait databases like TRY, 6]. Low-cost monitoring and sensing solutions have also immensely increased the amount of data individual researchers can produce [e.g., 7]. However, the data deluge—often from heterogeneous sources—introduces new logistical and computational challenges for researchers [7, 8] wanting to maintain best practices in data analyses, reproducibility and transparency [see frameworks on workflow implementation in 9, 10]. It is clear that we need not only frameworks, but also flexible tools to deal with the ever-increasing, heterogeneous data and corresponding issues.

Paramount to any analyses is ensuring the validity of input data through adequate exploration and quality control, which allows identifying any idiosyncrasies, outliers or erroneous structures. However, with growing data volumes this becomes increasingly difficult. Indeed, several definitions establish "big data" at the threshold where single entities (i.e. researchers, institutions, disciplines) are no longer able to manage and process a given data set due to its size or complexity [e.g., 11, 12]. Yet, several current research applications in ecology and Earth system science require handling more than Gigabyte-scale data and regularly lead to the development of dedicated and domain-specific processing pipelines and tools, e.g., processing of raw data from FLUXNET [4] or automating data assimilation from long-term experiments [13].

Individual scientists, however, frequently encounter data sets smaller than this, which nonetheless challenge the feasibility of common data processing and exploration methods. These include the best practice examples of generating static diagnostic/summary visualizations, statistics and tables for detecting problematic observations [e.g., 14, 15]. Data sets of this intermediate scale are termed "high-volume," rather than "big," for purposes of this study. Issues with these data often arise when the dimensions and data types require numerous of the aforementioned items (e.g., n-dimensional visualizations), and their individual assessment becomes unfeasible due to time and practicality constraints, even when their generation can be largely automated. Hence, they can pose a challenge even for experienced researchers adept at script-based analyses, if convenient tools do not exist or are financially inaccessible due to commercial licensing. For instance, over-plotting may require generating several static visualizations for nested categorical levels, such as branch, individual tree and forest stand, or for spatial granularity, such as plot, site and region. Furthermore, time series from monitoring equipment may show issues related to sensor drift, step-shifts, or random sensor errors. While gap-filtering, trend-adjustment and outlier-removal algorithms exist for these circumstances [e.g., 16, 17], subsequent manual checking is usually still advised, leading to similar issues as above. For time series, in particular, problematic periods (e.g., systematically-occurring sensor errors) may be removed entirely for convenience in code-based processing; by contrast, interactive engagement down to individual observations may allow applying more diligence and retaining more data.

Ideally, researchers should be able to engage with their data, across scales and dimensions as diligently as needed, with as little effort as possible. Accordingly, interactive processing is increasingly called for and deemed critical [18] for ensuring best practices in data exploration, and quality control when dealing with high-volume data and beyond [e.g., 19, 20]. Indeed, interactive exploration is increasingly provided through open-source graphing frameworks (e.g., plotly; https://plotly.com/ or, highcharts; https://highcharts.com/) and/or commercially-

licensed software (e.g., Tableau®; https://tableau.com/). However, actual data manipulation, and especially the generation of subsequent outputs that are fully reproducible, are far less common features; this could potentially stimulate reluctance for sharing analysis code [e.g., 21]. Further issues can arise when outputs are (commercially licensed) platform/software dependent and thus not easily incorporated with other widely-used languages, such as R [22] or python [23]. Interactive, reproducible is, therefore, typically linked to method-specific workflows within research domains, for instance, to annotate images [e.g., 24], acoustic files [e.g., 25], or explore spatial and time series data [e.g., 20, 26].

There is a clear need for interactive tools that can facilitate best practices in processing heterogeneous, high-volume data, while enabling interoperability with reproducible workflows. To address this, we developed datacleanr: an open-source R-based package containing a graphical user interface for rapid data exploration and filtering, as well as interactive visualization and annotation of various data types, including spatial (georeferenced) and time series observations. datacleanr is designed to fit in existing, scripted processing (R) pipelines, without sacrificing the benefits of interactivity: this is achieved through features that allow validating the results of previous quality control, and by generating a code script to repeat any interactive operation. The code script can be slotted into existing workflows, and datacleanr's output can hence be directly used for subsequent reproducible analyses.

Below we provide an overview of the package. Additionally, we demonstrate datacleanr's utility with two ecology-based use-cases addressing common issues during data processing: 1) Identifying problematic data structures and artefacts using an urban tree survey, where data is nested by species, street and city district. 2) Preventing excessive loss of data from "coarse," code-based filtering in messy time series of sap flow data, bolstering subsequent analyses.

Lastly, we provide an outlook for future developments and conclude by inviting the community to contribute to further increase datacleanr's capabilities and reach.

## Datacleanr overview

### Availability

This publication used v1.0.3 of datacleanr, which is permanently archived on Zenodo under https://doi.org/10.5281/zenodo.6337609. Stable package releases are available on the Comprehensive R Archive Network (CRAN; use install.packages("datacleanr")), which aim to mirror new developments provided and documented on a dedicated repository (www.github.com/the-hull/datacleanr). The repository provides installation instructions for all sources (CRAN, repository) and animated demonstrations with test data. datacleanr is available under a GPL-3 license.

### Capabilities

datacleanr is an interactive R package for processing high-volume data, and it caters to best practices in data exploration, processing, and reproducibility. This section describes the general capabilities of the package, and an in-depth walk-through of all functionalities is provided with animated examples in S1 File in the supplemental material. The package uses the shiny [27] and plotly [28] packages to generate a web browser-based graphical user interface (GUI), where modern browser capabilities allow displaying approximately 2 million observations smoothly, around which the visualizations and processing increasingly slow down (dependent on computing power). The GUI has four modules represented in application tabs, which are documented using intuitively-placed help links and package documentation: 1) Set-up and Overview (grouping and exploration), 2) Filtering, 3) Visual Cleaning and Annotating, 4) Extraction (reproducible recipe). The processing GUI is launched with datacleanr::dcr_app(x)

in R, where x is a data set for processing (several data types are possible, including data.frames, tibbles and data.tables; run? datacleanr::dcr_app() for help).

The chart in Fig 1 shows the datacleanr workflow across the four modules (A-D) with optional pre- and post-processing with external algorithms. Users are encouraged to cycle through multiple grouping structures, filters and variable combinations to get adequately acquainted with their data. The functions of individual tabs are discussed in detail below.

**Set-up and overview.** datacleanr facilitates processing of nested data through defining a grouping structure (Fig 1A; also see animation at https://doi.org/10.5281/zenodo.6469658) to the level of interest (e.g., by selecting species, plot and region). The structure is available during targeted filtering (scoping; see section *Filtering*) and visual cleaning (see section *Visual cleaning and annotating*, as well as *Case studies*). Once the grouping is set, a dataset summary can be generated via the package summarytools [30], highlighting duplicates, missingness, and distribution of each variable.

**Filtering.** Filtering (Fig 1B; also see animation at https://doi.org/10.5281/zenodo.6469721) is done by adding filter statement text boxes by clicking on the respective button on the tab. Statements can be applied to the entire data set, or targeted to specific groups using a "scoped" (i.e. group-specific) filter. The application's interactivity allows reviewing the impact of filters through a text note highlighting the percentage of removed data and an overview table showing the remaining observations (per group), as well as by iterating between settings and visualizations (across several variable combinations). This is more efficient than (re-)generating individual, static figures, and highlights which data will be excluded. The result of a quantile-based threshold filter implemented in datacleanr, as used e.g., in TRY [6] or BAAD [31], is illustrated in Fig 2.

Example of the impact of statistical filtering on bivariate relationships between trait data from BAAD [31]. A percentile threshold filter (0.01 and 0.99) is used to remove extreme low and high values on the x-variable across its full space (left) or scoped to groups represented by functional types (right). The gray shading indicates the filtered variable space (full or scoped), while text labels and black points count and highlight, respectively, individual observations captured by the applied filter. Note, the figure was not generated in datacleanr.

Any filtering statement (provided as valid R code) which evaluates to TRUE or FALSE and using the dataset's column names can be used, as shown below. That is, let example_numeric_-variable and example_categorical_variable be column names, then following statements are valid and can be supplied in a filter statement box:

```
# simple logical filter
example_numeric_variable > 3
# using expressions to define thresholds
## percentile/rank based
example_numeric_variable > quantile(example_numeric_variable, 0.01)
## dispersion based (median absolute distance)
example_numeric_variable > median(example_numeric_variable)-
3 * mad(example_numeric_variable)
# example for subsetting
example_categorical_variable = = "SpeciesA"
example_categorical_variable %in% c("SpeciesA", "SpeciesB")
```

**Visual cleaning and annotating.** Interactive visualizations (Fig 1C; also see animation at https://doi.org/10.5281/zenodo.6469756) via plotly [28] are based on bivariate scatter or time series plots with optional third dimension represented by point size. Spatial data can be displayed on interactive map tiles, if columns named lon (longitude) and lat (latitude) are present in decimal degrees. Example visualizations for currently supported data types are in Fig 3.

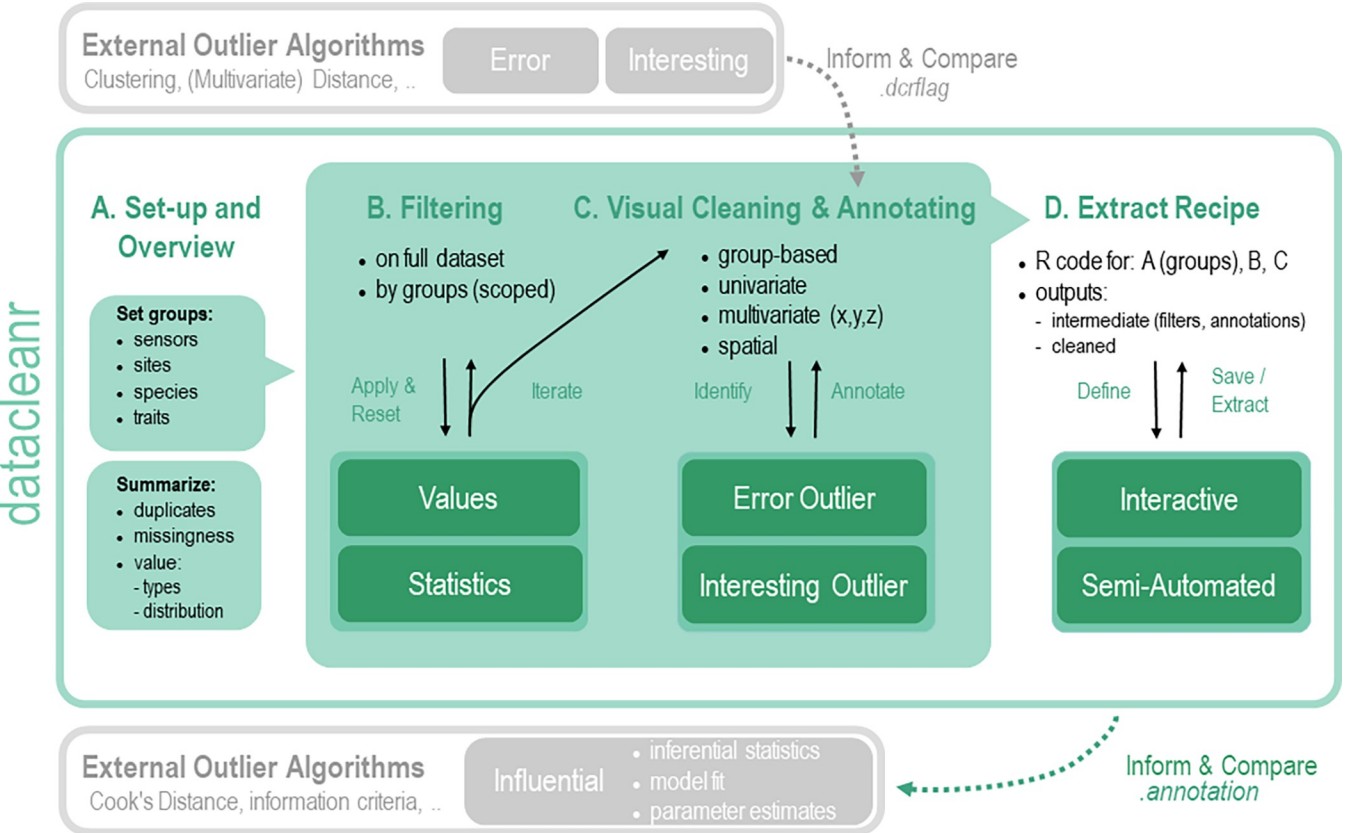

**Fig 1. Conceptual workflow for datacleanr across its four processing modules.** A) The Set-up and Overview tab allows for a quick initial assessment of a data set (variable types, distribution, completeness), where nested structures (e.g., by plot, site, region) can be resolved by defining a grouping structure from a categorical data column. B) The Filtering tab allows sub-setting the data based on valid R code (logical statements), which can be targeted (i.e., "scoped") to individual groups from A). C) The Visual Cleaning and Annotating tab allows generating two or three dimensional visualizations (X, Y, and point size) rapidly, while dividing the data set into groups specified in A); data points for further inspection can be identified by clicking or lasso selection through which annotations may also be added. An overview table and histogram highlight selected points and the potential impact on the data's distribution, should the selected observations be removed. D) The Extract Recipe tab generates code to reproduce all processing steps, which can be copied to the clipboard or sent directly to an active RStudio® [29] session's script; depending on the processing mode (in memory or from a file), additional settings for file name specification are available. The schematic here illustrates the potential for including datacleanr into an existing workflow, for example, with prior determination of outliers using external algorithms (requires appending a logical TRUE/FALSE column named.dcrflag), interactive exploration and processing (with datacleanr), and informing subsequent analyses by drawing on the interactively annotated data (.annotation column in output from datacleanr).

A key feature on the visualization tab is the grouping structure table, which allows highlighting granular data levels, e.g., all species at a given site, by hiding all other data (see Figs 7 and 8). Entries in this table correspond to colored traces in the figure legend via unique numbers. Users can thus cycle through or compare deeply nested data structures. Visualizations support zooming and panning (scatter plots and maps), as well as axes scrolling and stretching on mouse hover-over. Observations can be (de-)selected through clicking or lasso and box selecting, and annotated with text labels, which are listed in a summary table below the visualization. Annotations can be provided in a text box, and are added either individually through a button click, or automatically on every selection (requires ticking corresponding box); these are added to the input data in an appended column (.annotation) and can be used to inform subsequent processing. Lastly, histograms of all displayed, numeric variables can be generated to assess the potential impact of data removal.

**Extract recipe.** Reproducibility requires that any analyses step can be recovered, comprehended, and executed identically, repeatedly, and independently of the user. The datacleanr

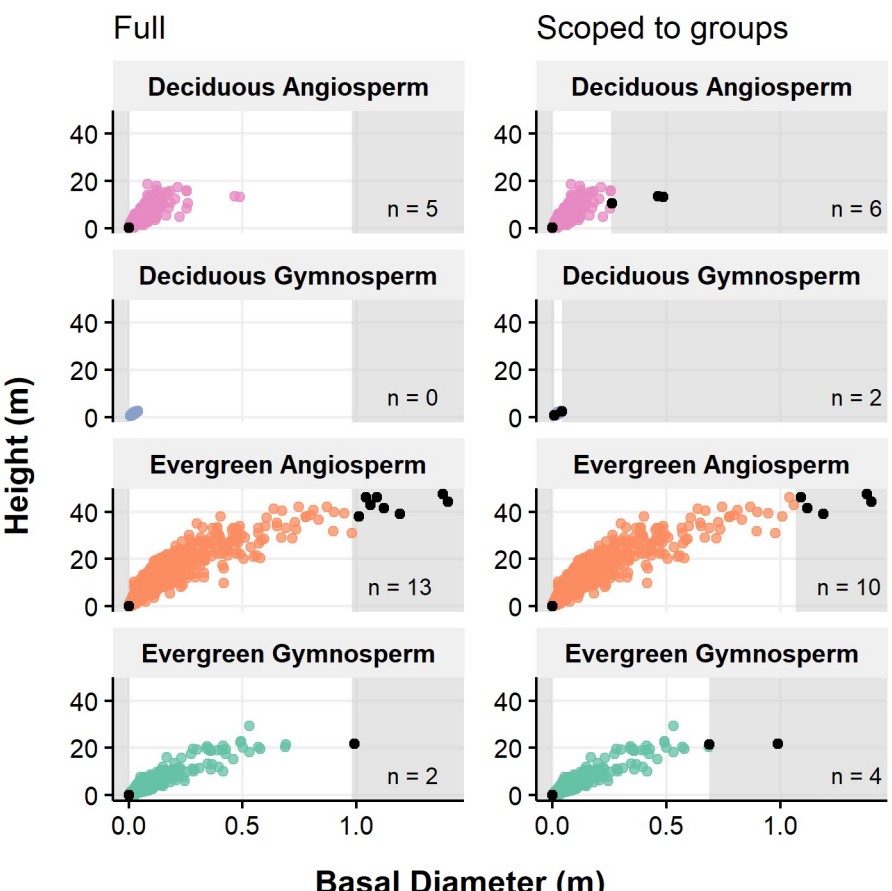

**Fig 2. Example application of statistical filtering.** A percentile threshold filter (0.01 and 0.99) is applied to the full range or scoped to groups (panels, see bold text) on the x-variable. Trait data from BAAD [31] is used to illustrate the impact on bivariate relationships across plant functional types, where the gray shading indicates the filtered variable space and text labels count the excluded observations (e.g.," n = 2"). Note, the figure was not generated in datacleanr.

package caters to this by translating every processing (filtering, highlighting or annotating) action into R code on the Extract Recipe tab, which can be copied or directly sent to an active RStudio® [R development environment, 29] session (Fig 1D; also see animation at https://doi.org/10.5281/zenodo.6469767). This code represents a recipe to reproduce the interactive processing, and survives the interactive session; subsequent analyses steps can thus include and build on the recipe (i.e., code script) for generating quality-controlled data. The dcr_app can also be launched with a file path to an *.RDS file on disk, rather than an object in R's environment (i.e., memory). In this case, additional saving options are available for adjusting output names and file locations (Fig 4). This is currently recommended for data requiring extensive annotation, which would result in code scripts of considerable length. However, the option will be made available for both modes (file path, object) in a future version.

**Interoperability with external packages and algorithms.** Interoperability is achieved with pre- and post-processing (in R) by two means:.

1. Observations that were flagged by prior outlier or data processing have distinct symbols in interactive visualizations; this is enabled by adding (or renaming) a logical (TRUE / FALSE) flagging column named.dcrflag before launching dcr_app().

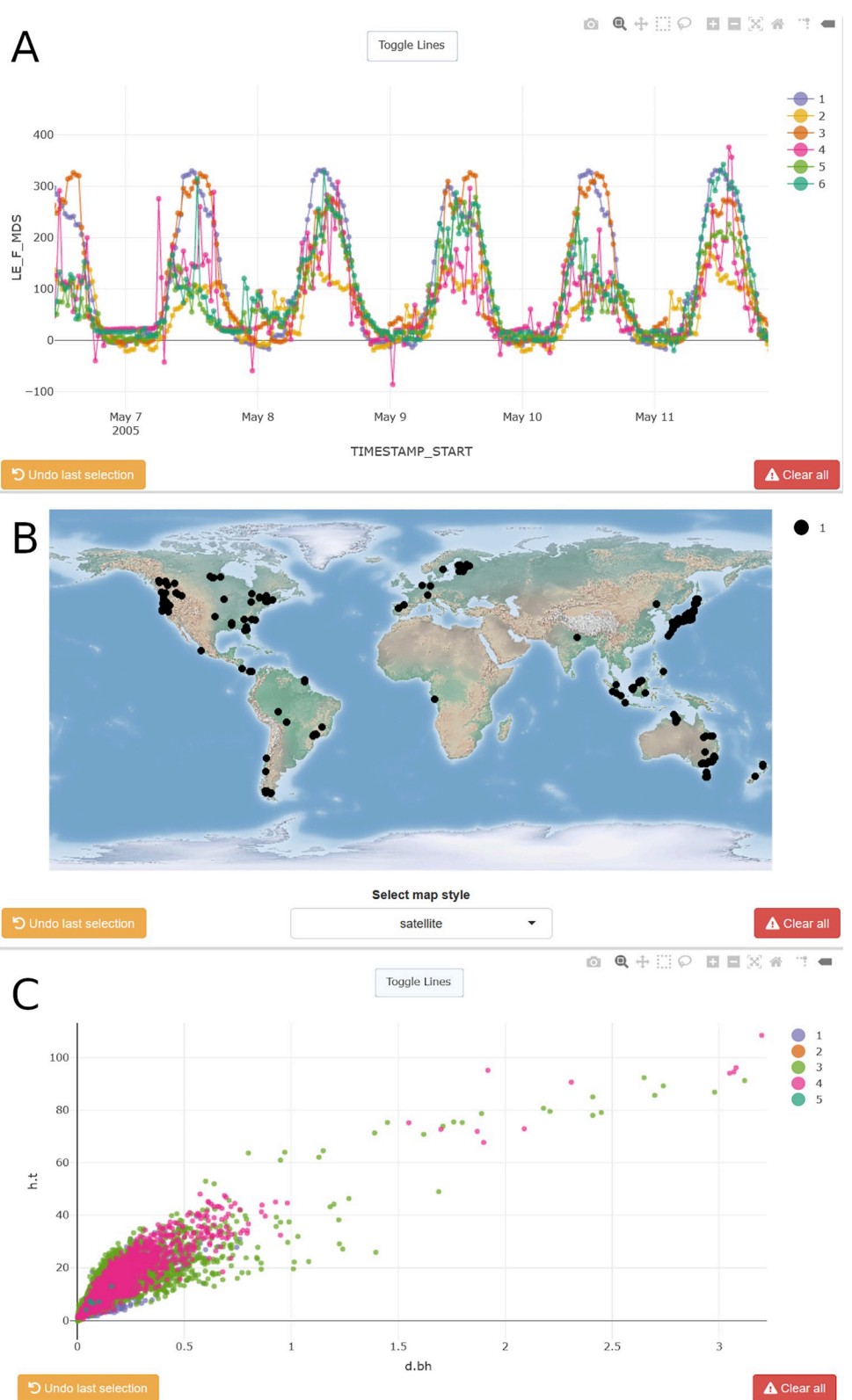

**Fig 3. Examples of interactive visualizations.** Panels show a subset of hourly time series of latent heat fluxes (A) from all Swiss FLUXNET2015 sites [4], spatial data illustrating sample locations for BAAD (B) and the relationship between stem diameter and height (C) with plant traits from BAAD [31]. Colors represent the grouping structure defined in the "Set-up" operation (A: Swiss sites; B, C: functional types).

2. The reproducible code recipe can be used as a step following or preceding additional analyses.

datacleanr can hence be embedded in R- based workflows with the existing strengths of R's extensive ecosystem of packages to increase flexibility and, ultimately, productivity. For instance, a script-based workflow applying widely-used R packages for reading, "wrangling" and cleaning data, such as readr [32], dplyr [33], tidyr [34] and lubridate [35] from the tidyverse [36], as well as janitor [37], can be complemented with datacleanr's interactivity. In addition, due to the script and file-based output, datacleanr can also be included in workflow management tools such as drake [38] and workflowr [39].

## Efficiency and batching

datacleanr has been extensively tested on mobile and desktop workstations (Windows 10 and 11, Ubuntu 19.10) considered medium to high-end, and is easily capable of processing and displaying above 1 million observations simultaneously. A speed test with outlier selections at excessive and improbable scales indicated comfortable response times for most user scenarios (Fig 5).

Nevertheless, the notable limiting factors for processing are number of columns (in exploratory summary) and the grouping structure during plotting. That is, a large number of unique groups will slow down the visualization, and we recommended aiming for a maximum of around 100 groups per datacleanr run.

datacleanr::dcr_app() returns processed results to the active R session. Hence, multiple datasets can be processed in batch and results (including code) saved for subsequent use. This is especially helpful when data nesting structures are too deep (e.g., ≫100 groups), or datasets too large (approximately above 2 million observations) to handle in one sitting. In these cases, a split-combine approach is recommended:

```
# prepare data into species sub-sets
iris_split <- split(x = iris,
          f = iris$Species)
# run for each species
dcr_iris <- lapply(iris_split,
          function(split){
                datacleanr::dcr_app(split)
          })
```

Similarly, a list of file paths to datasets can be supplied (see help in R via? datacleanr::dcr_app()).

## Case studies

Below are two use-cases illustrating the utility of the interactive approach adopted in datacleanr.

### Identifying structure and artefacts in nested data

High-volume, data with nested hierarchical structure (i.e. observations grouped at many levels) are difficult to explore and process, especially when obtained from secondary sources, where the data-generating process or the collection method are not fully known [e.g., see 41]. In such cases greater care is required, as unexpected or erroneous structures and artifacts can be present—especially so when these vary at group-level. Here, interactively engaging with the data can expedite processing while increasing confidence. As an example of dealing with such scenarios in datacleanr, we present a subset of nearly 320000 city and park trees listed in

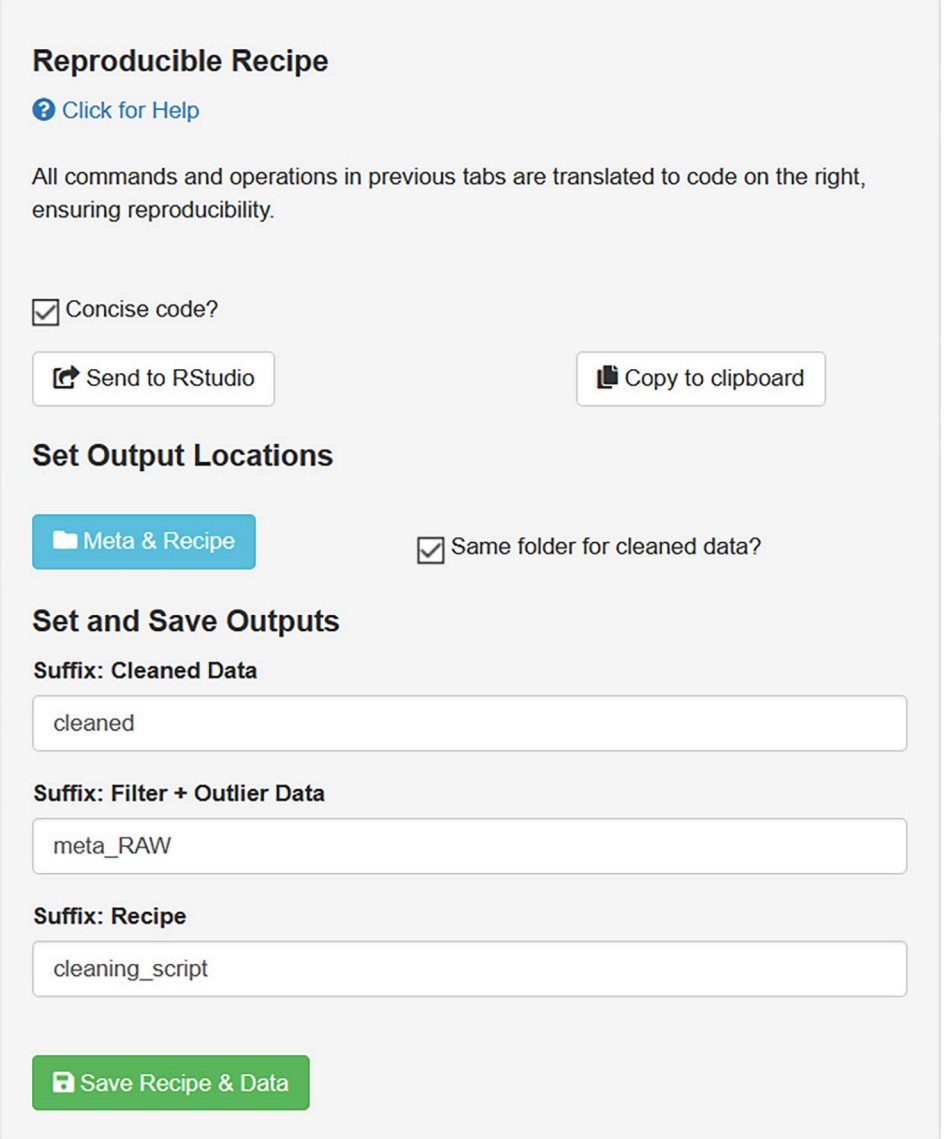

**Fig 4. Code recipe extraction.** Options provided in the Extract Recipe tab for defining and saving outputs when datacleanr is launched with a file path. Copying or sending the recipe (i.e. R code) to an active RStudio® [29] session is always possible.

Berlin's (DE) green infrastructure registry (https://daten.berlin.de/, Strassenbäume, Anlagenbäume), focusing on mensuration data from the 10 most-frequent species across all 12 districts. These data were collected by different agencies or contractors (within and between districts), and are nested at multiple levels (district, street/park). For convenient exploration with datacleanr, the data is grouped by district and species (Fig 6), giving 120 sub-groups.

Potential outliers are readily identified and annotated in a bivariate plot of tree age and diameter (Fig 7). Note, these observations could also be captured using threshold filters.

Upon cycling through the set groups, however, additional structures are apparent in the district of Neukölln for *Quercus robur* L (Fig 8), among others. These structures would only be apparent if individual visualizations (at least 120, and potentially at variable zoom) had been

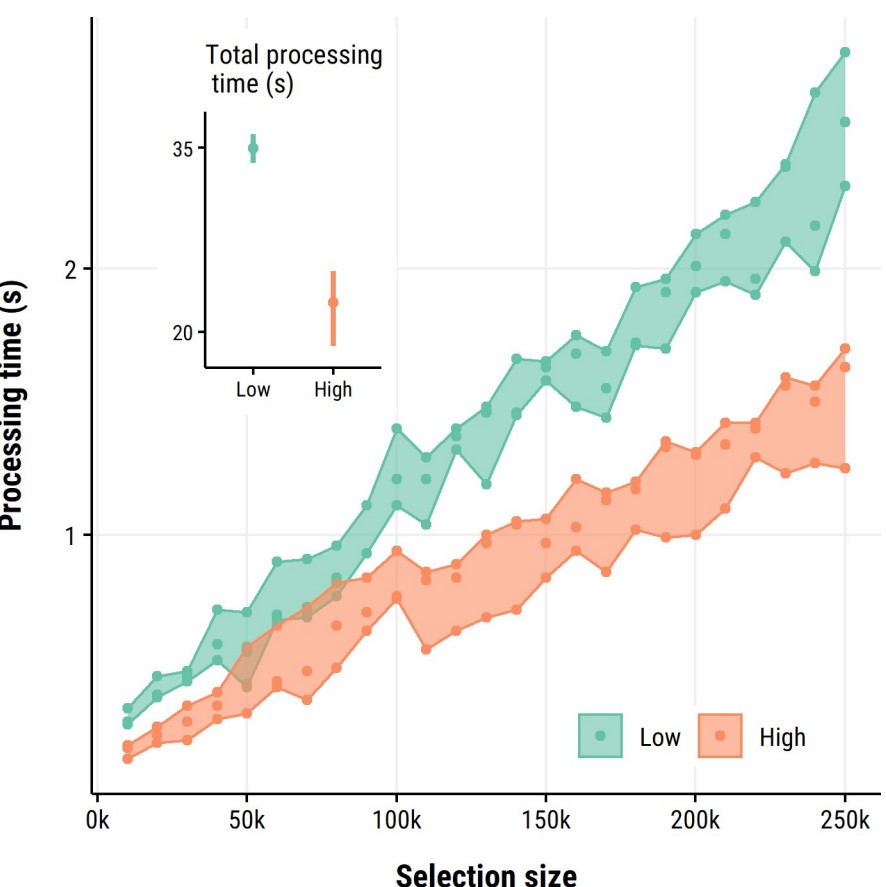

**Fig 5. Speed test of visualization and data selection on synthetic data (n = 250000).** In 25 consecutive steps 10000 (additional) observations were selected. This was repeated three times (points) on low and high CPU-power settings, and processing time was determined using profvis [40], with bands plotted as visual aids. The inset shows processing totals (mean, min, max) after completing all 25 selections. Even with selections representing unlikely outlier numbers, the application remained responsive and appreciably fast.

generated. Yet, they could not be removed easily with threshold filters and would require a high level of effort to address with manual or automated code-based processing.

Contrastingly, with scroll and zoom in datacleanr, problematic observations are efficiently selected and annotated. Such observations could be erroneous, and, for example, pose an issue in hierarchical modeling, if an entire group structure is affected (e.g., random effect at park or street level). We take this opportunity to explicitly urge users to make extensive use of the annotation feature on the visualization tab to provide rich information on the selected observations (e.g., "interpolated observations"), as well as to adhere to best practices and transparency in outlier assessment and handling [e.g., 42] for any subsequent removal.

## Retaining more data from time series with interactive cleaning

Time series data, e.g., from ecophysiological monitoring, can be messy due to instrument drift, response lags, power issues, etc. In high volumes, messy data may call for pragmatic decisions, such as indiscriminately removing entire periods, if interactive processing tools are not available. Such decisions may be owed to either time constraints for detailed manual processing, or because automated approaches may not accommodate unexpected processes and resulting observations. Interactive processing—both after automated quality control as well as in first

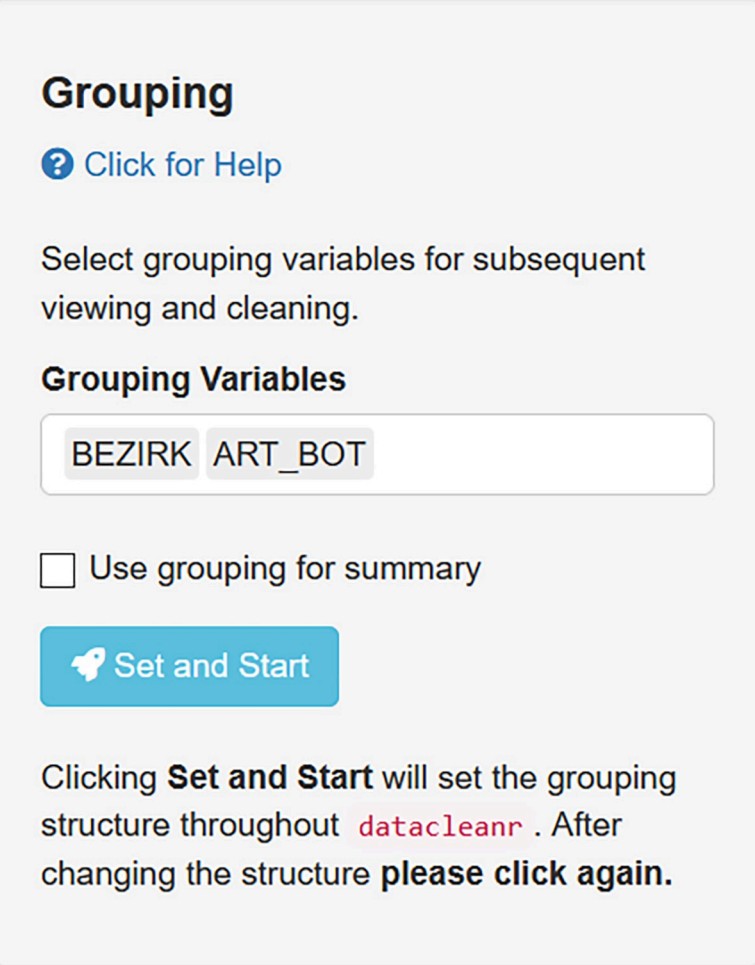

**Fig 6. Example set-up for hierarchically nested data.** Grouping structure set to district (BEZIRK) and species (ART_BOT) in "Set-up and Overview" tab for subsequent exploration, plotting and cleaning.

instance—with datacleanr allows inspecting high-volume data at high resolution and to identify the impact of erroneous data points. Consequently, individual problematic observations, rather than entire periods, can be flagged and removed after careful consideration. We provide an example of manual (code-based period filtering) *vs.* interactive processing with datacleanr of an unpublished (in prep.) time series of raw sap flow data from the TERENO North-East Observatory [Müritz National Park, 43]. Note, in both cases due diligence and best practices were applied. With datacleanr more observations were retained, as individual points—not only periods—could be removed. Consequently, resulting gaps were shorter, could be gap-filled and potentially provide a higher level of insight (Fig 9). Further, the processing time decreased from approximately 2 hours by a skilled R-user to under 15 minutes for the entire series.

## Future developments

Development will be continued to enhance performance, and incorporate user feedback. Additional improvements are planned and will be implemented in upcoming versions. These include: 1) saving and loading processing progress within the application, 2) pre-select groups

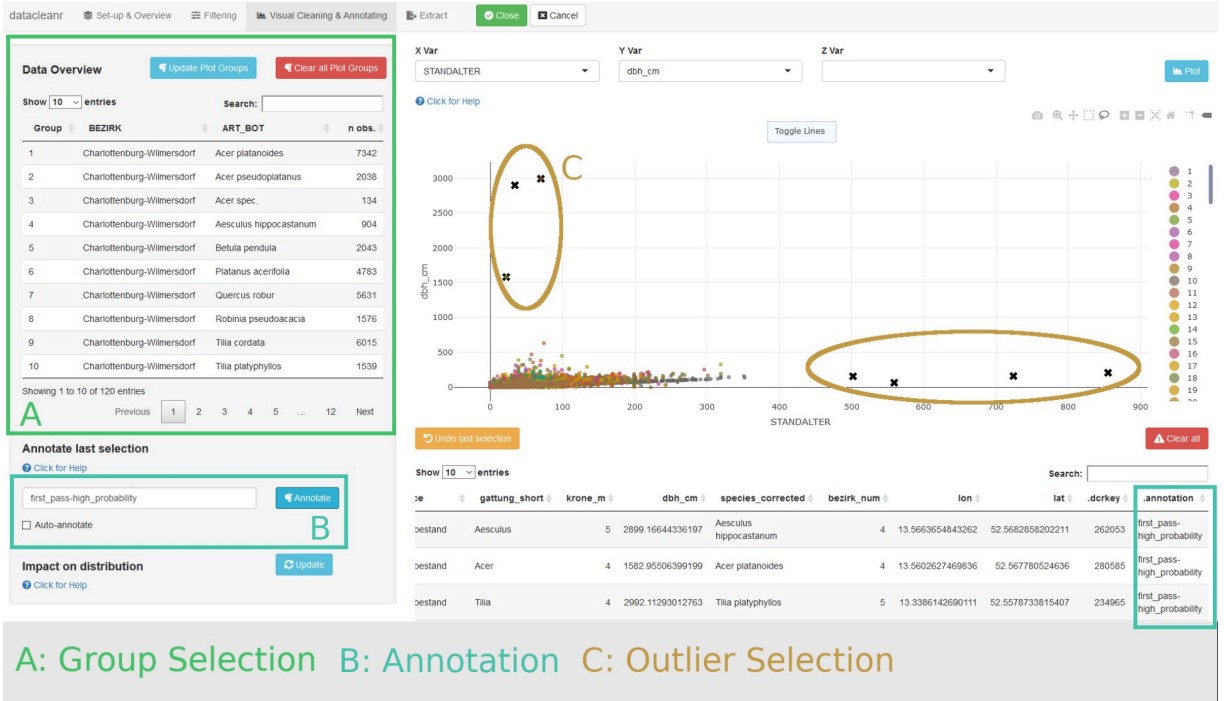

**Fig 7. Overview of visual cleaning tab for Berlin tree data.** The plot shows tree age (x) and diameter (y), resolving nesting by district and species. All 120 groups are displayed (see A, and figure legend). Potential outliers are obvious and easily highlighted and annotated for later reference (B, C). The dense point cloud comprises nearly 320000 observations and requires further inspection.

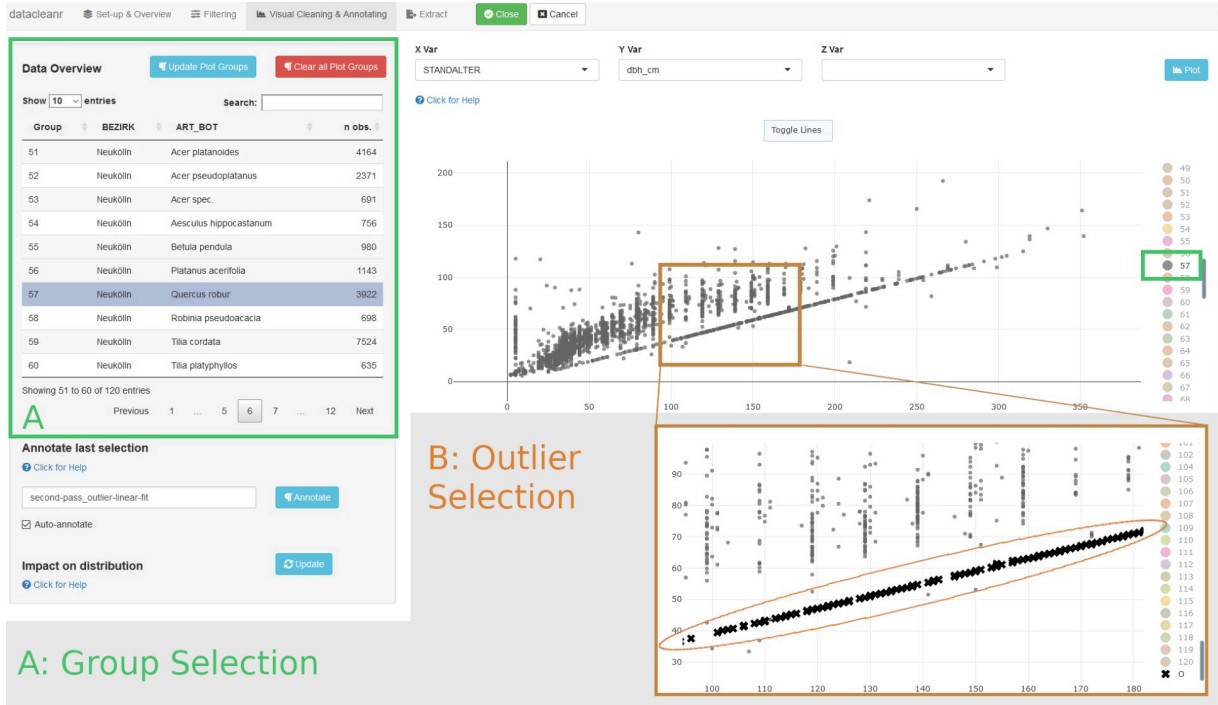

**Fig 8. Identification of problematic data structures.** Closer inspection of the tree dataset using the grouping structure (A) to highlight/hide specific groups. The obvious, near-perfect linear relationship between tree age and diameter at breast height requires further inspection. Concerning data points are easily selected leveraging the interactivity of the visualization by clicking or with a lasso tool (B, see inset).

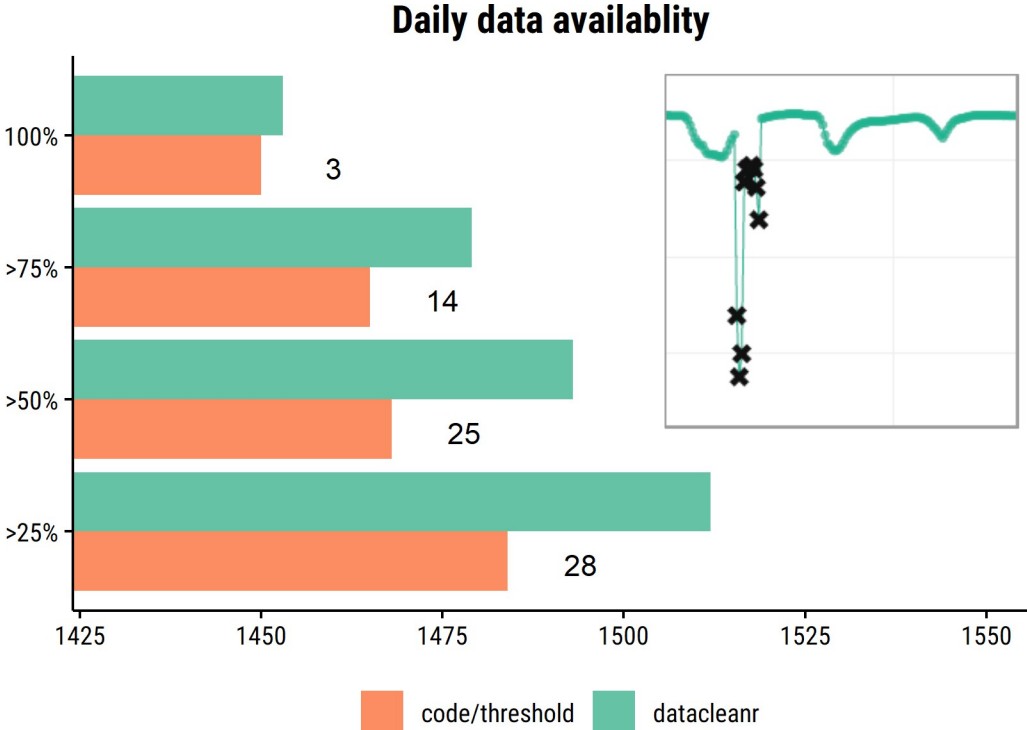

**Fig 9. Comparison of code-based and interactive processing with datacleanr of raw sap flux data.** Compared to often more tedious, code-based filtering, the interactive quality control using domain expertise allowed retaining more observations resulting in greater data coverage across days (x-axis) for the measurement campaign, which lasted a total of 2769 days. Here, three additional full days of measurements, as well as several additional days with varying partial coverage were retained, as indicated by text labels to the right of bars (completeness by day; e.g. 25% of all measurements for a given day). This is because individual, problematic observations could be removed interactively (see inset), which may increase explanatory power in subsequent analyses.

for plotting to reduce loading times, 3) a toggle to display filtered data (from the Filter tab) in visualizations for easier assessment of filters. Further, 4) a more convenient method for gracefully handling data selections from multiple groups in the interactive visualization will be added; this is particularly helpful when, for instance, problematic observations cluster in similar plot regions. Lastly, 5) additional options for data input via data base connections and internal splitting of (large) data sets will be added.

## Conclusion

Exploration and processing of high-volume data can be enhanced by using interactive tools. datacleanr achieves this with its flexibility and interoperability, while facilitating best practices in data exploration, outlier detection, and especially reproducibility through the extractable code recipe. While we acknowledge the place for and utility of fully-automated processing pipelines, we are certain that freely-available, interactive tools will improve researchers' and analysts' necessary engagement with their data, and consequently, increase confidence in their results. Further, we believe the datacleanr's design will increase productivity of both technically-proficient as well as users with limited programming ability. For this, we ensured it would fit seamlessly into existing, script-based analyses pipelines, or that it could be used as a stand-alone tool by a wide audience. Lastly, we hope datacleanr will be of great use to the scientific community, including ecology, Earth system sciences and fields working with spatial and

temporal data in general. We encourage users to provide feedback and suggestions in the dedicated repository to drive the continued development of the application.

## Supporting information

**S1 File. Animated walk-through.** An overview of the package's functionalities with animated examples of every feature.
(HTML)

## Acknowledgments

We are grateful for feedback on early versions of this manuscript and the application by anonymous reviewers.

## Author Contributions

**Conceptualization:** Alexander G. Hurley.

**Data curation:** Alexander G. Hurley, Richard L. Peters, David N. Steger, Ingo Heinrich.

**Funding acquisition:** Ingo Heinrich.

**Investigation:** Richard L. Peters, David N. Steger, Ingo Heinrich.

**Software:** Alexander G. Hurley, Richard L. Peters, Christoforos Pappas.

**Validation:** Alexander G. Hurley, Richard L. Peters, Christoforos Pappas, David N. Steger.

**Visualization:** Alexander G. Hurley.

**Writing – original draft:** Alexander G. Hurley.

**Writing – review & editing:** Alexander G. Hurley, Richard L. Peters, Christoforos Pappas, David N. Steger, Ingo Heinrich.

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
