## [Decision Letter · Decision Letter 0]

15 Feb 2022

PONE-D-21-05607Addressing the need for interactive, efficient and reproducible data processing in ecology with the datacleanr R applicationPLOS ONE

Dear Dr. Hurley,

Thank you for submitting your manuscript to PLOS ONE. After careful consideration, we feel that it has merit but does not fully meet PLOS ONE’s publication criteria as it currently stands. Therefore, we invite you to submit a revised version of the manuscript that addresses the points raised during the review process.

The reviewers raised a number of concerns with your study, in particular the ease of use of the program and the lack of a walk-through analysis, as well as some points to improve clarity. Their comments can be viewed in full, below and in the attached file.

We look forward to receiving your revised manuscript.

Kind regards,

Natasha McDonald, PhD

Associate Editor

PLOS ONE

Journal Requirements:

2. Your abstract cannot contain citations. Please only include citations in the body text of the manuscript, and ensure that they remain in ascending numerical order on first mention.

3. We note that Figure 3 in your submission contain map images which may be copyrighted. All PLOS content is published under the Creative Commons Attribution License (CC BY 4.0), which means that the manuscript, images, and Supporting Information files will be freely available online, and any third party is permitted to access, download, copy, distribute, and use these materials in any way, even commercially, with proper attribution. For these reasons, we cannot publish previously copyrighted maps or satellite images created using proprietary data, such as Google software (Google Maps, Street View, and Earth). For more information, see our copyright guidelines: http://journals.plos.org/plosone/s/licenses-and-copyright.

a. You may seek permission from the original copyright holder of Figure 3 to publish the content specifically under the CC BY 4.0 license.  

4. We note that Figures 4, 6, 7 and 8 in your submission contain copyrighted images. All PLOS content is published under the Creative Commons Attribution License (CC BY 4.0), which means that the manuscript, images, and Supporting Information files will be freely available online, and any third party is permitted to access, download, copy, distribute, and use these materials in any way, even commercially, with proper attribution. For more information, see our copyright guidelines: http://journals.plos.org/plosone/s/licenses-and-copyright.

a. You may seek permission from the original copyright holder of Figures 4, 6, 7 and 8 to publish the content specifically under the CC BY 4.0 license. 

Reviewers' comments:

Reviewer's Responses to Questions

**Comments to the Author**

1. Is the manuscript technically sound, and do the data support the conclusions?

Reviewer #1: Yes

Reviewer #2: Yes

2. Has the statistical analysis been performed appropriately and rigorously? 

Reviewer #1: N/A

Reviewer #2: N/A

3. Have the authors made all data underlying the findings in their manuscript fully available?

Reviewer #1: Yes

Reviewer #2: Yes

4. Is the manuscript presented in an intelligible fashion and written in standard English?

Reviewer #1: Yes

Reviewer #2: Yes

5. Review Comments to the Author

Reviewer #1: Addressing the need for interactive, efficient and reproducible data processing in ecology with the dataclearnr R application

The manuscript introduces an R package for data processing of large ecological data sets. It gives an overview of the package's functionality.

I think that such an R package is useful, especially for the scenarios the authors have pointed out (large data sets with many different scales / scopes, large monitoring data sets), for quick visualizations, outlier and problem detection.

I only have a few suggestions for improving the manuscript:

1. line 139 Figure 1 caption: Sometimes the authors refer to specific functionality that the reader would only really understand when running the package, e.g. line 149 after "Set and Start" is clicked. This detail is perhaps too technical for this overview.

2. line 151 via 29 .... would be more useful if it named the package directly

3. line 156: not sure what 'through text cues on the tab' refers to

4. Figure 2 and caption: To me it is not entirely clear what is displayed here. E.g. what is the difference between left and right, what the n = .. refer to? Make clearer that the grey areas correspond to points which are filtered out?'

5. line 167 --- 'any statement': make clearer that this refers to the filtering statement, it would be useful to add that this filtering statement requires R code

6. line 169: insert 'the' before 'following'

7. line 185: more useful to refer directly to 'plotly'

8. line 186: insert 'be' before 'displayed'

9. line 197: clarify what 'key feature' refers to. I think it refers to the visualization tab, but this is not clear.

10. line 199: correspond to

11. line 242, Figure 5 caption:not sure what 'processing totals' refer to. Why (n=3) after means? Does this imply that there were 3 runs at every setting (number of points)?

12. line 270: remove comma after high-volume

13. Figure 9 and caption (line 331): I cannot see a difference between figures A and B. Therefore, is A necessary. If there are differences, maybe highlight these or point out. Also, I am not sure how to understand panel C. 'including' in caption misplaced?

Reviewer #2: The paper describes an R package which provides an interactive graphical user interface to identify outliers and clean the data.

It has one extremely important feature, i.e. it returns the R code for doing the filtering and the cleaning of the data, i.e. therefore making it transparent and reproducible. This is an essential feature, which bridges the gap between purely code based outlier removal and interactive outlier identification.

But as the R script is effectively a script adding a column identifying if a data point was identified as an outlier or not, it would be very useful to also generate a report which includes e.g. the graphs in which the outliers were identified and the filtering rules as an html or pdf which can be added to the data as a justification why the points were identified as outliers.

In the ideal case (future development?) I would suggest a config file (yaml?) which contains all the info and settings used, and when loaded, loads the data and applies all settings from the previous session.

Unfortunately, I was only able to test the app after quite a bit of trying, as no walk-through of the data analysis is provided. This is a pity, as all the data is available in appropriate licenses and all that would be needed is to supplement the manuscript with boxes showing the settings for each step. This relatively easy addition would make the manuscript much more approachable.

Overall, I have added a number of comments to the pdf document (attached) which can be easily adressed.

As mentioned, my main concern is the missing of a walk-through through one analysis. I would suggest that this relatively straight forward addition / change is done before publication.

6. PLOS authors have the option to publish the peer review history of their article (what does this mean?). If published, this will include your full peer review and any attached files.

Reviewer #1: No

Reviewer #2: No

---

## [Author Response · Author response to Decision Letter 0]

1 Apr 2022

Dear Editor and referees,

Thank you for the chance to improve our manuscript; your comments were highly appreciated. I kindly refer you to the attached file "Response to Reviewers.docx" for a formatted version of our rebuttal, but have copy-pasted it below for completeness.

Sincerely,

Alexander Hurley

______

Addressing the need for interactive, efficient, and reproducible data processing in ecology with the datacleanr R package

Response to reviewers

Editor

The reviewers raised a number of concerns with your study, in particular the ease of use of the program and the lack of a walk-through analysis, as well as some points to improve clarity. Their comments can be viewed in full, below and in the attached file.

We thank the editor and reviewers for their time and the opportunity to improve the manuscript and thus the (initial) user experience of datacleanr. We have carefully considered the concerns raised by the referees and included a detailed walk-through analysis to further enhance the usability of the presented package. With these suggestions we feel that the manuscript has significantly improved. 

Reviewer 1

The manuscript introduces an R package for data processing of large ecological data sets. It gives an overview of the package’s functionality. I think that such an R package is useful, especially for the scenarios the authors have pointed out (large data sets with many different scales / scopes, large monitoring data sets), for quick visualizations, outlier and problem detection.

We thank the reviewer for their assessment and recognition of our tool’s utility.

I only have a few suggestions for improving the manuscript:

We have addressed all comments by implementing the suggested changes or by providing a justification for the issue at hand in its current state. An overview thereof is given below.

*1. line 139 Figure 1 caption: Sometimes the authors refer to specific functionality that the reader would only really understand when running the package, e.g. line 149 after “Set and Start” is clicked. This detail is perhaps too technical for this overview.

We improved the wording of technical details here, enhanced Fig 1 caption (package overview, L165 in response) to better guide the reader, and added a SI File with animated examples for a walk-through, as also suggested by Reviewer 2. The section mentioned above now reads:

“The structure is available during targeted filtering (scoping; see Filtering) and visual cleaning (see Figs 6-8). Once the grouping is set, a dataset summary can be generated via the package summarytools [29], highlighting duplicates, missingness, and distribution of each variable.”

As noted above, the caption of Fig 1 has been improved significantly, in accordance with Reviewer 2’s suggestions (see in respective section further below, L165), which we believe allows the reader to follow the descriptions more readily.

*2. line 151 via 29 …. would be more useful if it named the package directly

Adjusted to “..summary can be generated via the package summarytools [29]..”

*3. line 156: not sure what ‘through text cues on the tab’ refers to

The section in question now reads:

“The application’s interactivity allows reviewing the impact of filters through a text note highlighting the percentage of removed data and an overview table showing the remaining observations (per group), …”

*4. Figure 2 and caption: To me it is not entirely clear what is displayed here. E.g. what is the difference between left and right, what the n = .. refer to? Make clearer that the grey areas correspond to points which are filtered out?’

We appreciate the caption was not detailed enough and have adjusted it to:

“Figure 2: Example of the impact of statistical filtering on bivariate relationships between trait data from BAAD [30]. A percentile threshold filter (0.01 and 0.99) is used to remove extreme low and high values on the x-variable across its full space (left) or scoped to groups represented by functional types (right). The gray shading indicates the filtered variable space (full or scoped), while text labels and black points count and highlight, respectively, individual observations captured by the applied filter. Note, the figure was not generated in datacleanr.”

We are confident that the adjusted caption sufficiently explains the figure and highlights the difference between full and scoped filtering adequately.

*5. line 167 — ‘any statement’: make clearer that this refers to the filtering statement, it would be useful to add that this filtering statement requires R code

The text was adjusted to:

“Any filtering statement (provided as valid R code) which evaluates to TRUE or FALSE..”

*6. line 169: insert ‘the’ before ‘following’

Done.

*7. line 185: more useful to refer directly to ‘plotly’

Done.

*8. line 186: insert ‘be’ before ‘displayed’

Done.

*9. line 197: clarify what ‘key feature’ refers to. I think it refers to the visualization tab, but this is not clear.

This was changed to:

“A key feature on the visualization tab is the grouping structure table…”

*10. line 199: correspond to

Done.

*11. line 242, Figure 5 caption: not sure what ‘processing totals’ refer to. Why (n=3) after means? Does this imply that there were 3 runs at every setting (number of points)?

The caption was adjusted to:

“Figure 5: Speed test of visualization and outlier selection on synthetic data (n = 250000). In 25 consecutive steps 10000 (additional) points were selected. This was repeated three times on low and high CPU-power settings, and processing time was determined using profvis [35]. Bands represent minimum and maximum durations, and points are means of the three replicates. The inset shows processing totals (mean, min, max) after completing all 25 selections. Even with unlikely outlier numbers, the application remained responsive and appreciably fast.”

*12. line 270: remove comma after high-volume

Done.

*13. Figure 9 and caption (line 331): I cannot see a difference between figures A and B. Therefore, is A necessary. If there are differences, maybe highlight these or point out. Also, I am not sure how to understand panel C. ‘including’ in caption misplaced?

We appreciate this issue, and have previously tried different versions of the figure with overplotting and offsetting in a single panel. However, neither option was fully satisfactory. This was either due to the same issue (distance between lines) or overplotting due to the time scale. We do want to emphasize the entire time series to highlight the package’s capability of dealing with fairly large/high-resolution data here for one processing example, and have used the caption to better highlight visible differences, , although these admittedly require increased attention. The caption now reads:

“Figure 9: Comparison of code-based (A) and interactive processing (B) of raw sap flux data. Compared to often more tedious, code-based filtering, interactive quality control using domain expertise allowed retaining more observations resulting in greater data coverage. Here, three additional full days of measurements, as well as several days with varying partial coverage were retained (C, completeness by day; e.g. 25 % of all measurements for a given day). For example, differences are found in 2014 and 2017, where the code-based filtering removes entire periods (i.e., days, weeks). By contrast, individual, problematic observations could be removed interactively (D), which may increase explanatory power in subsequent analyses.”

Reviewer 2

*It has one extremely important feature, i.e. it returns the R code for doing the filtering and the cleaning of the data, i.e. therefore making it transparent and reproducible. This is an essential feature, which bridges the gap between purely code based outlier removal and interactive outlier identification.

We thank the reviewer for the recognition of datacleanr’s utility and the thorough review.

*But as the R script is effectively a script adding a column identifying if a data point was identified as an outlier or not, it would be very useful to also generate a report which includes e.g. the graphs in which the outliers were identified and the filtering rules as an html or pdf which can be added to the data as a justification why the points were identified as outliers.

We discussed similar features for PDF or HTML reports during datacleanr’s development. We agree that a justification or annotation for conspicuous data is not only useful but necessary. This is why we implemented the annotation feature in the interactive data selection. This gives users the freedom to use a set of self-defined annotations or tags for different scenarios (e.g., “high value,” “battery failure,” etc.). These annotations are stored in the .annotation column, and allow to not only identify selected points in a boolean manner, but also provide the user-specified annotation. From our own use of datacleanr we have concluded that this approach affords flexibility down the line to implement case-specific solutions, such as bespoke graphs and tables, which can be included in user-specific outputs (e.g., PDF or HTML reports based on RMarkdown). We also would like to emphasize that the reproducible recipe provides all the necessary information for such reports - or even as a stand-alone overview - if the data selection and annotation is done with due diligence.

*In the ideal case (future development?) I would suggest a config file (yaml?) which contains all the info and settings used, and when loaded, loads the data and applies all settings from the previous session.

This is most certainly planned as a future development, and will likely rely on the new shiny caching feature, rather than a yaml config file – we will explore the latter option as well and appreciate the idea/pointer, however. As we recognize the importance of this feature it is also the first we mention in this section (with updated wording):

“Additional improvements are planned and will be implemented in upcoming versions. These include: 1) saving and loading processing progress within the application …”

*Unfortunately, I was only able to test the app after quite a bit of trying, as no walk-through of the data analysis is provided. This is a pity, as all the data is available in appropriate licenses and all that would be needed is to supplement the manuscript with boxes showing the settings for each step. This relatively easy addition would make the manuscript much more approachable.

Thank you for highlighting this. We have included the package’s readme file (https://github.com/the-Hull/datacleanr or on CRAN as https://cran.r-project.org/web/packages/datacleanr/readme/README.html) with animated examples as an SI file (modified to comply with the 20 MB file restriction), which we consider even more instructive than a walk-through with screenshots only. It includes details on installation, capabilities, and use, with in-depth examples of every feature. We note this in the “Capabilities section” to prime the reader. The paragraph now reads:

“datacleanr is an interactive R package for processing high-volume data, and it caters to best practices in data exploration, processing, and reproducibility. This section describes the general capabilities of the package, and an in-depth walk-through of all functionalities is provided with animated examples in S1 File).”

*Overall, I have added a number of comments to the pdf document (attached) which can be easily adressed.

Thank you for the thorough comments. We have adjusted the text accordingly.

A list of noteworthy alterations or responses to comments in the PDF beyond simple text adjustments:

 -the title of the article was changed to read “package” rather than “application,” and is referred to as such throughout the article now

 -added DOI (https://doi.org/10.5281/zenodo.6337609) and version number (v1.0.3) to “Availability” section

 -Updated data availability statement to include “latest version” Zenodo DOI, for which the archive now contains CSV data only: https://doi.org/10.5281/zenodo.4550726

 -instead of noting an explicit example with respect to financially inaccessible software, we rephrased to “..if convenient tools do not exist or are financially inaccessible due to commercial licensing”

 -tibbles can be supplied to dcr_app(); the help documentation for dcr_app() notes that the data can be a data.frame, tbl (tibble), or data.table

 -The caption for Figure 1 was modified to address all processing modules. It now reads: Figure 1: Conceptual workflow for datacleanr across its four processing modules. A) The Set-up and Overview tab allows for a quick initial assessment of a data set (variable types, distribution, completeness), where nested structures (e.g., by plot, site, region) can be resolved by defining a grouping structure from a categorical data column. B) The Filtering tab allows sub-setting the data based on valid R code (logical statements), which can be targeted (i.e., “scoped”) to individual groups from A). C) The Visual Cleaning and Annotating tab allows generating two or three dimensional visualizations (X, Y, and point size) rapidly, while dividing the data set into groups specified in A); data points for further inspection can be identified by clicking or lasso selection through which annotations may also be added. An overview table and histogram highlight selected points and the potential impact on the data’s distribution should the selected observations be removed. D) The Extract recipe tab generates code to reproduce all processing steps, which can be copied to the clipboard or sent directly to an active RStudio session’s script; depending on the processing mode (in memory or from a file), additional settings for file name specification are available. The schematic here illustrates the potential for including datacleanr into an existing workflow, for example, with prior determination of outliers using external algorithms (requires appending a logical TRUE/FALSE column named .dcrflag), interactive exploration and processing (with datacleanr), and informing subsequent analyses by drawing on the interactively annotated data (.annotation column in output from datacleanr).

 -We considered your comment on using text-based data objects, as opposed to binary files. We strongly feel that the binary format for use with datacleanr is necessary, as this is the only way to ensure that data input/output does not alter data types, for example from factor or time to character, which may require that additional bespoke code is added to the extracted recipe. We cannot ensure that this is would be done correctly during generation of the recipe, and thus prefer the *.RDS format. We appreciate the limitation on universal data input/output, however.

 - Is the cleaning (or any other aspects in this app) paralellised, and could you gain a substantial increase if you do this?

Currently, the data selection (point clicking and lasso selection) is implemented by carrying point indices and plotly trace numbers (i.e., group numbers) in a data.frame. The limiting factors are checking for duplicates and redrawing the “outlier” trace (which is plotly’s equivalent of a ggplot2 geom). In fact, due to plotlys mechanics, the entire plot needs to be redrawn or refreshed when traces are manipulated. Packages such as multidplyr may be interesting for subsetting individual groups in the Filtering tab, but those operations are certainly not a bottleneck in processing time. We do, however, strive to enhance the user experience in the future by further decreasing processing time with streamlining above mentioned data.frame operations.

 -We appreciate you highlighting the necessity to provide information on why observations were considered conspicuous, erroneous or outliers, and have rephrased line 310 (PDF) to: We take this opportunity to explicitly urge users to make extensive use of the annotation feature on the Visualization tab to provide rich information on the selected observations (e.g., “interpolated observations”), as well as to adhere to best practices and transparency in outlier assessment and handling [e.g., 37] for any subsequent removal.

 -Comments on future developments were:

 + maybe included but not mentioned here: generate report which can include comments to why certain decisions were taken -THIS WOULD BE A VERY USEFUL FEATURE. See previous comments on annotation tool and flexibility (L111 in response)

 + loading from other file formats csv, txt, databases (DBI). See previous comments on file formats; database connections were data types are preserved could be a viable option, however, which we will consider for future developments

 + parallelisation of processing. See previous comments (L199 in response)

 + include splicing in the app (probably loading data into an sqlite database and query subsets out)? We have included this in the list of future developments within the text (L401 in Revised Mansuscript with Track Changes.docx)

 + I don’t know if it is possible - reduce the dependencies. We have done our best to keep dependencies as low as possible, but shiny and plotly are rather heavy. However, we still strive to drop dependencies where possible, and will aim to do so in the future, e.g., for new developments.

 + add vignettes to the package (possibly this paper? License?). We appreciate the utility of vignettes. However, the GitHub repository (https://github.com/the-Hull/datacleanr) has an extensive set of examples with GIFs, which we believe are better suited to highlight the package’s functionality – this now constitutes the walk-through in the supporting information in a somewhat reduced fashion to meet the <20 MB file size requirements We also would like to note that CRAN very regularly enforces a <5 MB package size policy, and believe the GIFs in the ReadMe do better justice to the packages functionalities than a screenshot-based document of smaller size. The license for the package is GPL-3 and listed in the Availablility section.

 + In response to the comment on enhancement of data analyses through interactivity, we rephrased the first sentences of the conclusion to: “Exploration and processing of high-volume data can be enhanced by using interactive tools. datacleanr achieves this with its flexibility and interoperability, while facilitating best practices in data exploration, outlier detection, and especially reproducibility through the extractable code recipe.”

*As mentioned, my main concern is the missing of a walk-through through one analysis. I would suggest that this relatively straight forward addition / change is done before publication.

We appreciate this concern and are confident our animated examples in the Supporting Information file S1 File (Walkthrough.html) sufficiently address this concern.

---

## [Editor Report · Decision Letter 1]

13 Apr 2022

PONE-D-21-05607R1

Addressing the need for interactive, efficient, and reproducible data processing in ecology with the datacleanr R package

PLOS ONE

Dear Dr. Hurley,

I took over the editorial role for your manuscript to PLOS ONE and would like to acknowledge my role as Reviewer 2 in the last round of review.

I acknowledge the extremely long time since your initial submission but I was only assigned the role as editor for this paper a few days ago and will do anything possible to bring this paper to publication as soon as possible. A few comments are listed below in this letter.

Thank you for submitting your manuscript to PLOS ONE. After careful consideration, we feel that it has merit but does not fully meet PLOS ONE’s publication criteria as it currently stands. Therefore, we invite you to submit a revised version of the manuscript that addresses the points raised during the review process.

Thanks a lot for addressing all points raised by the reviewers. I have only a few minor points which should be adressed before publication and some more general comments:

Text:

Please include a reference for RStudio, as it is done for e.g. RFigure 2 caption: clarify what n meansSort figs according to their first occurrence in the manuscript (i.e. 6-8 after fig 1)Availability Section:Start with stating which version you are using in the package and mention the DOI - than mention stable release and where one can get these from (DOI to newest version, 10.5281/zenodo.6337608, CRAN) and finally github repo as newest version. Probably one sentence about release plans to CRAN (always latest stable), github master / main is stable? Do you have a dev branch for development and possibly unstable versions?Capabilities l 135 add "in the supplemental material" or "LINK TO THE PERMANENT FILE IN GITHUB"l 238: "2 million observations smoothly" - is this a hard link, or soft link, after which it get's slower but still work? Also, in line 270 you mention one milion.I stil think it would be nice to have screenshots in the sections "Capabilities" when you refer to the individual steps (you have them in the visualization section), but I appreciate your point that the animated gifs give a better point. Can you put direct links to the animated gifs in the section they refer to and keep these permanent (e.g. link to a specific tagged release on github)?l 264: Add individual references to the individual packages (readr, dplyr, tidyr and lubridate)

Figures:

Figure 5: with three replicates, to plot mean,min and max (which are two of the three points) is an aggressive approach. I would rather leave the shaded area and plot, instead of the mean, the third point. I do not see it as necessary as in this context, as the graph is not at all crucial to the paper, but regard would regard it as a cleaner approach.Figure 9 caption: re comment reviewer 1 on difference: I agree - I still, although explained in the caption, struggle to see where there is a difference. Also, I think the actual values of deltaV are not relevant in this graph. I think as a y asis, you would simply need four categories (from top to bottom): data in original data, day removed by using code based filering, day removed by using interactive processing, days gained by using interactive processing. These would be the relevant info, as the values at these days are not relevant based on the caption. If you want to retain the graph as it is, I would strongly suggest to have a fourth (or fifth?) colour, which highlights the data points retained in addition to the code based filtering.

SI:

I like the SI a lot and the gifs work brilliantly - thanks.

SI: Example 2 has missing GIF

General comments which do not require any action from your side:

text based data object versus RDS: most data is stored in a csv file, as it can be generated from e.g. Excel. So using these as input would be very useful. Export of results does not need to be lossless, ut only include relevant results from the result and additional info could be saved as text files with additional details or even rds files. Saving should be lossless, so RDS is here appropriate. CRAN issues - use RUniverse for "full" package, provide functions in the package to download the additional info and data when needed, ..l 269: If you put the shiny app on shiny server, you can use it from all platforms (even smart phones) which have a browser. Probably include this in future plans (not relevant to the paper here).Reading in from databases (sqlite and duckdb come to mind as widely used stand alone databases which are used for larger datasets when standard handling in R is not possible anymore) would be extremely useful as a next step.

We look forward to receiving your revised manuscript.

Kind regards,

Rainer M Krug, PhD

Academic Editor

PLOS ONE
---

## [Author Response · Author response to Decision Letter 1]

21 Apr 2022

Dear Editor, 

We extend our thanks for your time (also for the previous review) and the opportunity to further improve the manuscript. We have carefully considered the concerns and are confident that we fully address them in this second round of revisions. We kindly refer to the submitted response letter for detailed responses. Note that line references there refer to Revised Manuscript with Track Changes.docx

Sincerely,

Alexander Hurley on behalf of all authors

---

## [Editor Report · Decision Letter 2]

2 May 2022

Addressing the need for interactive, efficient, and reproducible data processing in ecology with the datacleanr R package

PONE-D-21-05607R2

Dear Dr. Hurley,

I received your revised version today and I am happy with the changes you made.

We’re pleased to inform you that your manuscript has been judged scientifically suitable for publication and will be formally accepted for publication once it meets all outstanding technical requirements.

Kind regards,

Rainer M Krug, PhD

Guest Editor

PLOS ONE

---

## [Editor Report · Acceptance letter]

5 May 2022

PONE-D-21-05607R2 

Addressing the need for interactive, efficient, and reproducible data processing in ecology with the datacleanr R package 

Dear Dr. Hurley:

I'm pleased to inform you that your manuscript has been deemed suitable for publication in PLOS ONE. Congratulations! Your manuscript is now with our production department. 

Kind regards, 

on behalf of

Dr. Rainer M Krug 

Guest Editor

PLOS ONE